# Visual routines for detecting causal interactions are tuned to motion direction

**Sven Ohl[1]\*, Martin Rolfs[1,2]**

[1]Department of Psychology, Humboldt-Universität zu Berlin, Rudower Chaussee, Berlin, Germany; [2]Berlin School of Mind and Brain, Berlin, Germany

## eLife Assessment

This study provides a **valuable** contribution to our understanding of causal inference in visual perception. The evidence provided through multiple well-designed psychophysical experiments is **convincing**. The current study targets very specific visual features of launch events, future work will be able to build on this to study the implementation of causal inference in general.

**Abstract** Detecting causal relations structures our perception of events in the world. Here, we determined for visual interactions whether generalized (i.e. feature-invariant) or specialized (i.e. feature-selective) visual routines underlie the perception of causality. To this end, we applied a visual adaptation protocol to assess the adaptability of specific features in classical launching events of simple geometric shapes. We asked observers to report whether they observed a launch or a pass in ambiguous test events (i.e. the overlap between two discs varied from trial to trial). After prolonged exposure to causal launch events (the adaptor) defined by a particular set of features (i.e. a particular motion direction, motion speed, or feature conjunction), observers were less likely to see causal launches in subsequent ambiguous test events than before adaptation. Crucially, adaptation was contingent on the causal impression in launches as demonstrated by a lack of adaptation in non-causal control events. We assessed whether this negative aftereffect transfers to test events with a new set of feature values that were not presented during adaptation. Processing in specialized (as opposed to generalized) visual routines predicts that the transfer of visual adaptation depends on the feature similarity of the adaptor and the test event. We show that the negative aftereffects do not transfer to unadapted launch directions but do transfer to launch events of different speeds. Finally, we used colored discs to assign distinct feature-based identities to the launching and the launched stimulus. We found that the adaptation transferred across colors if the test event had the same motion direction as the adaptor. In summary, visual adaptation allowed us to carve out a visual feature space underlying the perception of causality and revealed specialized visual routines that are tuned to a launch's motion direction.

## Introduction

When two objects collide, the objects' future path can be inferred based on physical laws, allowing us, for instance, to purposely change the motion path of an asteroid by steering an uncrewed spacecraft into a collision with the asteroid (see NASA's Double Asteroid Redirection Test; *Figure 1a*). In contrast, how humans detect such causal interactions in their visual environment is less clear. The proposed theories regarding our understanding of causal sensory interactions vary considerably, proposing purely perceptual (*Michotte, 1963*; *Scholl and Tremoulet, 2000*), abstract probabilistic (*Sanborn et al., 2013*), or cognitive mechanisms (*Rips, 2011*; *Weir, 1978*; *White, 2006*). These are neither mutually exclusive nor do they make distinct predictions and they are, therefore, hard to distinguish

\*For correspondence:
sven.ohl@hu-berlin.de

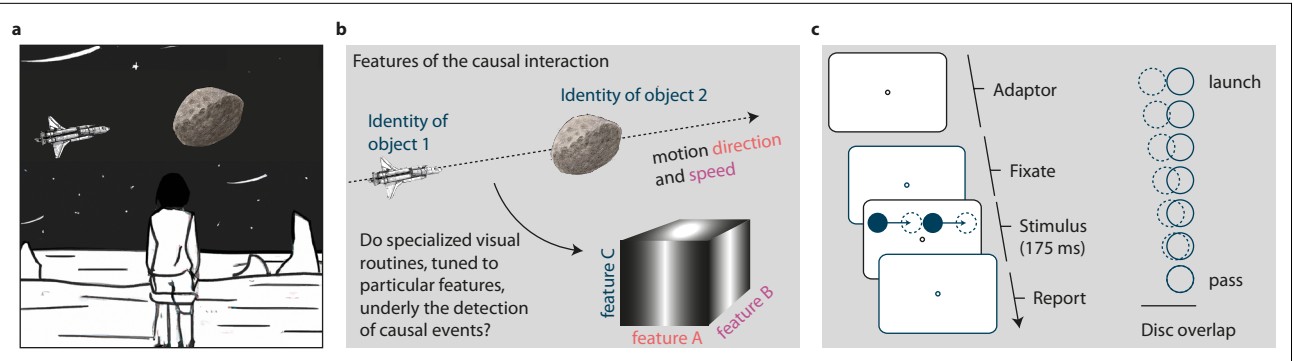

**Figure 1.** The perception of causality. (**a**) A person perceiving an upcoming causal interaction between an uncrewed spacecraft and an asteroid. (**b**) Features of such launching events are the direction and speed of the colliding objects as well as their object identities (i.e. a specific set of features). Assessing the adaptation's transfer between features allows us to determine whether the perception of causality arises in specialized visual routines that are tuned to a particular visual feature. (**c**) Trial sequence of a test event. A peripheral disc moved towards a stationary disc and stopped with some degree of overlap (ranging from 0–100% overlap in seven equidistant steps) between the two discs. The second disc then immediately started to move in the same direction, with the same speed as the first disc. In adaptation blocks, 320 launches were presented before the first test event of a block, and 16 top-up adaptation events before each subsequent test event.

(*Rips, 2011*). Here, we will study the mechanisms underlying the computations of causal interactions in the visual domain by capitalizing on visual adaptation of causality (*Kominsky and Scholl, 2020*; *Rolfs et al., 2013*). Adaptation is a powerful behavioral tool for discovering and dissecting a visual mechanism (*Kohn, 2007*; *Webster, 2015*) that provides an intriguing testing ground for the perceptual roots of causality. Perceptual accounts of causal understanding posit the existence of visual routines—local, semi-independent operations that can engage mid- and higher-level processes (*Ullman, 1987*; *Cavanagh et al., 2001*). The precise nature of visual routines that detect causal relations, however, remains rather hazy. In its purest form, such a visual routine could constitute a generalized mechanism that responds to all kinds of causal visual interactions. Alternatively, there could be many different specialized visual routines for the detection of causality that are tuned to the features of a particular causal interaction (e.g. the direction, kinematics, or object identity; *Figure 1b*; see *Rips, 2011* for a discussion). Here, we will use visual adaptation to determine whether the computation of causality occurs in generalized (i.e. feature-invariant) or in specialized (i.e. feature-selective) visual routines. Specifically, we will determine whether an adaptor is most effective for test events that match the features of the adaptor.

To this end, we took advantage of the phenomenon that observers perceive causality even in simple kinematic displays—a moving disc that stops next to another disc appears to launch the second disc into motion (i.e. launching effect, *Figure 1c*; *Michotte, 1963*; see *Rips, 2011*; *Scholl and Tremoulet, 2000*; *White, 2006*, *White, 2017* for reviews). The prolonged viewing of such launching events in an adaptation protocol strongly alters the perception of causality by reducing the proportion of reported launches in subsequent test events (*Rolfs et al., 2013*). The adaptation of causality is spatially specific to the retinotopic coordinates of the adapting stimulus (*Kominsky and Scholl, 2020*; *Rolfs et al., 2013*; for an object-centered elasticity aftereffect using a related stimulus on a circular motion path, see *Arnold et al., 2015*), suggesting that the detection of causal interactions is implemented locally in visual space. Here, we selected adaptors from a feature space—while keeping their spatial locations constant—and determined whether adaptation transfers to test events with a different set of features in that feature space (*Figure 1b*). If the strength of the adaptation's perceptual consequences depends on the similarity of the adaptor and the test event, it would reveal the existence of specialized visual routines for the detection of causal interactions that are tuned to that feature (i.e. there would be multiple visual routines within a feature dimension, each tuned to a different preferred feature value). Alternatively, if we observe adaptation transfer from one feature to another, this will support the notion of a generalized, feature-invariant, visual routine.

In three experiments, we assessed whether adaptation to launches of a particular motion direction, motion speed, or feature identity (i.e. a conjunction of two features) transfers to other values of that feature space or, alternatively, whether the consequences of adaptation are restricted to the feature

values of the adaptor. Our results provide compelling evidence for visual routines that are specialized for processing launches of a particular motion direction.

## Results

### Adaptation of causality is selective for motion direction

In **Experiment 1**, we tested whether causality is computed separately within visual routines specialized for different motion directions. To this end, we presented observers with brief test events in one of two possible horizontal directions (from left to right, or right to left) in which a peripheral disc moved swiftly towards a stationary one. The first disc stopped moving as soon as the two discs partially overlapped (the amount of overlap was manipulated in seven steps from zero to full overlap, *Figure 1c*). Simultaneously, the second disc started to move along the same trajectory, and we asked observers to report whether they perceived that the first disc passed the second one (i.e. a pass event; commonly reported at full overlap), or rather launched it into motion (i.e. a launch event; common at zero overlap). We quantified the point of subjective equality (PSE) between perceived launches and passes by modeling psychometric functions for the perception of causality both *before* and *after* adaptation to causal launches (note that we presented the adaptor in a range of ±30° around one of the two possible directions). Moreover, we compared the visual adaptation to launches to a (non-causal) control condition in which we presented slip events as adaptor. In a slip event, the initially moving disc passes completely over the stationary disc, stops immediately on the other side, and then the initially stationary disc begins to move in the same direction without delay. Thus, the two movements are presented consecutively without a temporal gap. This stimulus typically produces the impression of two independent (non-causal) movements.

Visual adaptation to launches successfully affected the perception of causality in this task. We observed a strong negative aftereffect when test events matched the direction of the launch adaptor. That is, observers were less likely to report a launch after adapting to launches in the same direction as the test events (*Figure 2a–c*). This observation was corroborated by a two-way (two event types: launch vs. slip; two directions: congruent vs. incongruent) repeated measures analysis of variance (rmANOVA) in which we assessed the magnitude of adaptation by subtracting the PSE before adaptation ($PSE_{before}$ = 0.60, $CI_{95\%}$ = [0.59 0.62]) from the PSEs obtained in each of the four different adaptation conditions ($PSE_{cong\_launch}$ = 0.40, $CI_{95\%}$ = [0.36 0.43]; $PSE_{incong\_launch}$ = 0.55, $CI_{95\%}$ = [0.54 0.56]; $PSE_{cong\_slip}$ = 0.55, $CI_{95\%}$ = [0.54 0.57]; $PSE_{incong\_slip}$ = 0.61, $CI_{95\%}$ = [0.59 0.64]). The analysis revealed a significant main effect of event type ($F_{(1, 7)}$=62.73, p<0.001; *Figure 2a–c*) demonstrating stronger adaptation for launching than slip events ($\Delta PSE_{event}$ = –0.11, $CI_{95\%}$ = [–0.14–0.08]). Moreover, adaptors in a congruent direction as the test event resulted in stronger adaptation than in incongruent directions ($F_{(1, 7)}$=41.62, p<0.001; $\Delta PSE_{direction}$ = –0.11, $CI_{95\%}$ = [–0.15–0.07]). Critically, we observed a significant interaction ($F_{(1, 7)}$=12.23, p=0.01), revealing that the influence of direction-congruency on the magnitude of adaptation was significantly stronger for launches than for slip events (Post-hoc t-test: $t_{(7)}$ = 3.50, p=0.01; $\Delta PSE_{eventXdirection}$ = –0.09, $CI_{95\%}$ = [–0.16–0.03]) supporting the hypothesis that specialized visual routines are tuned to the motion direction of launching events.

### Adaptation of causality transfers across motion speed

In **Experiment 2**, we determined whether the perception of causality is also selective for other feature dimensions that are relevant to causal interactions. Causal events in our visual environment encompass interactions of various motion speeds. Here, we examined whether adapting the perception of causality transfers between events that either had different speeds (same as in **Experiment 1** or half that). Both speed-congruent (i.e. test events had the same speed as the adaptor) and speed-incongruent launches (i.e. test events and adaptor differed by a factor of 2) resulted in adaptation, demonstrating a transfer of adaptation across motion speeds (*Figure 2d–f*). As in **Experiment 1**, we corroborated this observation by a two-way rmANOVA in which we assessed differences in the magnitude of adaptation (i.e. the difference in PSEs before and after adaptation; $PSE_{before}$ = 0.59, $CI_{95\%}$ = [0.57 0.61]; $PSE_{cong\_launch}$ = 0.43, $CI_{95\%}$ = [0.40 0.46]; $PSE_{incong\_launch}$ = 0.43, $CI_{95\%}$ = [0.39 0.47]; $PSE_{cong\_slip}$ = 0.60, $CI_{95\%}$ = [0.57 0.63]; $PSE_{incong\_slip}$ = 0.54, $CI_{95\%}$ = [0.50 0.59]) as a function of speed congruency (congruent vs. incongruent speed), event type (launch vs. slip event) and their interaction. The analysis revealed a significant main effect of event type ($F_{(1, 7)}$=23.39, p=0.002; *Figure 2d–f*) demonstrating

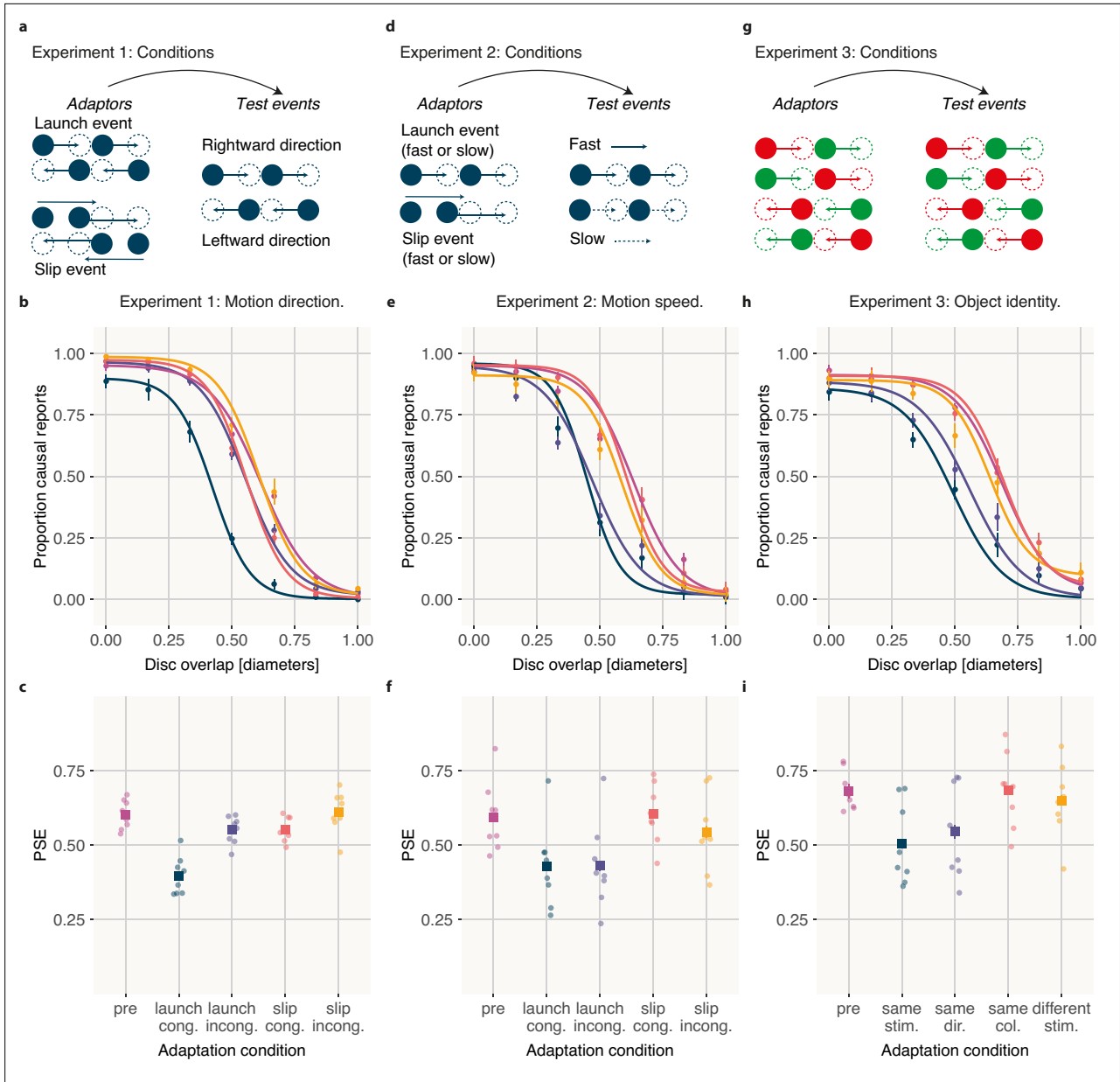

**Figure 2.** Results of Experiment 1, 2, and 3. (**a**) Experimental conditions of Experiment 1. (**b**) Mean proportion of causal reports as a function of disc overlap in Experiment 1. Visualization of psychometric curves is based on the mean parameters averaged across observers before adaptation (pink), and after adaptation with direction-congruent launches (blue), direction-incongruent launches (in purple), direction-congruent slip events (red) and direction-incongruent slip events (in orange). (**c**) PSEs for each individual observers (circles, n=8) and the mean across observers (square). (**d**) Experimental conditions of Experiment 2. (**e**) Mean proportion of causal reports as a function of disc overlap in Experiment 2. Visualization of psychometric curves is based on the mean parameters averaged across observers before adaptation (pink), and after adaptation with speed-congruent launches (blue), speed-incongruent launches (in purple), speed-congruent slip events (red) and speed-incongruent slip events (in orange). (**f**) PSEs for each individual observers (circles, n=8) and the mean across observers (square). (**g**) Experimental conditions of Experiment 3. (**h**) Mean proportion of causal reports as a function of disc overlap in Experiment 3. Visualization of psychometric curves is based on the mean parameters averaged across observers before adaptation (pink), and after adaptation with adaptors that are identical with the test event (blue), share the same direction but different colors (in purple), same colors but different direction (red) and completely different test events (in orange). (**i**) PSEs for each individual observers (circles, n=8) and the mean across observers (square). All error bars are ±1 SEM.

stronger adaptation for launching than slip events ($\Delta PSE_{event}$ = –0.14, $CI_{95\%}$ = [–0.21–0.07]). Importantly, the magnitude of adaptation was not significantly different for congruent and incongruent speed between adaptor and test event ($F_{(1, 7)}$=3.43, p=0.107; $\Delta PSE_{speed}$ = 0.03 $CI_{95\%}$ = [–0.01 0.07]). Moreover, there was no significant interaction between event type and speed ($F_{(1, 7)}$=3.48, p=0.105;

$\Delta PSE_{eventXspeed}$ = –0.06, $CI_{95\%}$ = [–0.14 0.02]). These findings demonstrate that the perception of causality is not tuned to motion speed—or, at least, that the tuning is very broad.

## Adaptation of causality using feature-conjunctions

In **Experiment 3**, we determined whether the identities of the two objects involved in the causal interaction can break the observed dominance of the direction-specific computation underlying the perception of causality. For instance, if two visually distinct objects are involved in a causal interaction with one type of object always launching a second, visually distinct object, it is unclear whether the influence of the adaptor is confined to events with the same feature conjunction or, alternatively, whether motion direction is the only relevant feature dimension for determining the presence of causal interactions. To distinguish between these alternatives, the two discs in the presented test events had two different colors (i.e. they were either red or green). Moreover, the adaptor in a session displayed only one particular feature conjunction (e.g. the adaptor in one session was always a red disc on the left launching a green disc on the right into rightward motion; adaptors varied across sessions and each subject saw each combination of feature conjunction across the multiple sessions). In contrast to the adaptor's fixed feature conjunction in a session, both motion direction and color identities in the test events varied randomly from trial to trial. Again, we observed strong adaptation when test events and adaptors were identical. However, when one of the features in the feature conjunction differed, we observed adaptation only for test events in the same direction as the adaptor, irrespective of the object's color. Thus, color does not constitute a critical feature of the visual routine for detecting causal interactions (*Figure 2g–i*). Again, we corroborated this observation by a two-way rmANOVA in which we assessed differences in the magnitude of adaptation (i.e. the difference in PSEs before and after adaptation; $PSE_{before}$ = 0.68, $CI_{95\%}$ = [0.63 0.73]; $PSE_{same\_stimulus}$ = 0.50, $CI_{95\%}$ = [0.47 0.53]; $PSE_{same\_direction}$ = 0.55, $CI_{95\%}$ = [0.50 0.59]; $PSE_{same\_color}$ = 0.68, $CI_{95\%}$ = [0.65 0.72]; $PSE_{different\_stimulus}$ = 0.65, $CI_{95\%}$ = [0.61 0.69]) as a function of motion direction (same vs. different), color assignment (same vs. different) and their interaction. The analysis revealed a significant main effect of motion direction ($F_{(1, 7)}$=21.07, p=0.003) demonstrating stronger adaptation for launches in the same as compared to the opposite direction as the adaptor ($\Delta PSE_{direction}$ = –0.14, $CI_{95\%}$ = [–0.22–0.07]). Adaptation following adaptors with the same color vs. different color assignments were not significantly different ($F_{(1, 7)}$=0.071, p=0.798; $\Delta PSE_{color}$ = –0.004, $CI_{95\%}$ = [–0.03 0.03]). However, we observed a marginally significant interaction between color and motion direction ($F_{(1, 7)}$=4.59, p=0.07; $\Delta PSE_{directionXcolor}$ = –0.07, $CI_{95\%}$ = [–0.16 0.01]). Post-hoc paired t-tests showed that color had a marginally significant influence on the magnitude of adaptation when adaptor and test events were in same direction ($t_{(7)}$=1.97, p=0.090; $\Delta PSE_{color\_samedir}$ = –0.04, $CI_{95\%}$ = [–0.09 0.01]). However, color had no significant influence on the adaptation of causality when adaptor and test events were in opposite direction ($t_{(7)}$=1.97, p=0.090; $\Delta PSE_{color\_opp\_dir}$ = 0.03, $CI_{95\%}$ = [–0.02 0.09]).

## Discussion

We provide new evidence for the fundamental role of perceptual processes in detecting cause and effect. Using visual adaptation, we revealed that visual routines underlie the perception of causality that are specialized for a launch's motion direction but invariant across a range of motion speeds. The tuning of causal perception to motion direction reveals a mechanism that is operating locally in feature space and complements previous reports of a spatially specific mechanism residing in retinotopic space (*Kominsky and Scholl, 2020*; *Rolfs et al., 2013*).

The observed adaptation is specific to launching events that evoked a phenomenological impression of causality. This implies that adapting simply to the direction of a moving stimulus will only change the input to the visual routine but not the functioning of the routine itself. We controlled for such potential non-causal adaptation of low-level features (e.g. number of stimuli, time of contact, speed, overlap, or simply motion) using carefully designed control events (i.e. slip events) that did not result in comparable aftereffects.

Motion direction constitutes a basic computational unit in vision and can be computed as early as in retinal circuits (e.g. in retinal ganglion cells about 2–3 synapses downstream of photoreceptors in rabbits, *Barlow and Hill, 1963*). While a retinal origin of launch detection is unlikely, the specialization of visual routines for the perception of causality at the level of individual motion directions

raises the possibility that this function is located surprisingly early in the visual system as opposed to a higher-level visual computation. A similar adaptation-based rationale can be applied to study whether different causal interactions can be distinguished from each other by assessing the transfer of adaptation between different causal interactions (*Kominsky and Scholl, 2020*). While Michotte speculated about three separate causal impressions (i.e. launching, entraining, triggering; see *White, 2017* for an extended catalog of causal interactions) adaptation helped to refine these categories as an adaptation to triggering events transferred to launching events, but adaptation to entraining had no such influence (*Kominsky and Scholl, 2020*). Thus, adaptation at the category level of the causal interaction allows to distinguish specialized detectors for a particular causal interaction. Our findings break down this specialization to a more fundamental level, showing that a specialized visual routine for launching events exists even within separate motion direction channels. While the present study demonstrates direction-selectivity for the detection of launches, previous adaptation protocols demonstrated successful adaptation using adaptors with random motion direction (*Rolfs et al., 2013*; *Kominsky and Scholl, 2020*). These results, therefore, suggest independent direction-specific routines, in which adaptation to launches in one direction does not counteract an adaptation to launches in the opposite direction (as for example in opponent color coding).

Habituation studies in 6-mo-old infants also demonstrated that the reversal of a launch resulted in a recovery from habituation to launches (while a non-causal control condition of delayed launches did not; *Leslie and Keeble, 1987*). In their study, the reversal of motion direction was accompanied by a reversal of the color assignment to the cause-effect-relationship. In contrast, our findings suggest, that in adults color does not play a major role in the detection of a launch. Future studies should further delineate similarities and differences obtained from adaptation studies in adults and habituation studies in children (e.g. *Kominsky et al., 2017*; *Kominsky and Scholl, 2020*).

The adaptation transferred across different speeds and colors as long as the test events shared the same motion direction as the adaptor. For instance, we demonstrated a transfer of adaptation across speed with symmetrical speed ratios. This result complements a previous finding that reported that the adaptation to triggering events (with a speed ratio of 1:3) resulted in significant retinotopic adaptation of ambiguous (launching) test events of different speed ratios (i.e. test events with a speed ratio of 1:1 and of 1:3; *Kominsky and Scholl, 2020*). Moreover, perceiving a red disc that launched a green disc into motion is also affected by the adaptation of a green disc launching a red disc (i.e. the opposite color assignment) into motion as long as the two events share the same motion direction. This finding rules out an alternative cognitive mechanism, in which the causing disc is first identified (e.g. the red disc is identified to launch other discs into motion) and the criterion for detecting a causal interaction would be selectively adjusted only for this particular causing disc in a top-down manner. Indeed, our results show that the appearance of the objects in the launching event plays only a minor role in adapting the perception of causality (if any), suggesting that the visual system can calibrate the detection of causal interactions independent of the involved object identities. Our findings, therefore, provide additional support for the claim that an event's spatiotemporal parameters mediate the perception of causality (*Michotte, 1963*; *Leslie, 1984*; *Scholl and Tremoulet, 2000*).

We suggest that at least two functional benefits result from a specialized visual routine for detecting causality. First, a direction-selective detection of launches allows adaptation to occur separately for each direction. That means that the visual system can automatically calibrate the sensitivity of these visual routines in response to real-world statistics. For instance, while falling objects drop vertically toward the ground, causal relations such as launches are common in horizontal directions moving along a stable ground. Second, we think that causal visual events are action-relevant, and the faster we can detect such causal interactions, the faster we can react to them. Direction-selective motion signals are available very early on in the visual system. Visual routines that are based on these direction-selective motion signals may enable faster detection. While our present findings demonstrate direction-selectivity, they do not pinpoint where exactly that visual routine is located. It is possible that the visual routine is located higher up in the visual system (or distributed across multiple levels), relying on a direction-selective population response as input.

The existence of specialized visual routines computing causality, however, does not rule out the possibility of a more complex hierarchical architecture that is composed of both specialized and generalized routines. The output of local, specialized detectors within that architecture could feed into a global, generalized detector. Thus, a weaker response from adapted specialized cause

detectors would elicit a weaker response in unadapted global detectors (or super-detectors; *Rips, 2011*). Indeed, visual adaptation to other low-level visual features (e.g. contrast sensitivity) can yield weaker responses during early visual processing which are then passed on to other downstream areas (*Kohn, 2007*).

We used visual adaptation to carve out a bottom-up visual routine for detecting causal interactions in form of launching events. However, we know that more complex behaviors of perceiving causal relations can result from integrating information across space (e.g. in causal capture; *Scholl and Nakayama, 2002*), across time (postdictive influence; *Cavanagh et al., 2001*), and across sensory modalities (*Sekuler et al., 1997*). Bayesian causal inference has been particularly successful as a normative framework to account for multisensory integration (*Körding et al., 2007*; *Shams and Beierholm, 2022*). In that framework, the evidence for a common-cause hypothesis competes with the evidence for an independent-causes hypothesis (*Shams and Beierholm, 2022*). The task in our experiments could be similarly formulated as two competing hypotheses for the second disc's movement (i.e. the movement was caused by the first disc vs. the second disc did not move). This framework also emphasizes the distributed nature of the neural implementation for solving such inferences, showing the contributions of parietal and frontal areas in addition to sensory processing (for review see *Shams and Beierholm, 2022*). Moreover, even visual adaptation to contrast in mouse primary visual cortex is influenced by top-down factors such as behavioral relevance—suggesting a complex implementation of the observed adaptation results (*Keller et al., 2017*). The present experiments, however, presented purely visual events that do not require an integration across processing domains. Thus, the outcome of our suggested visual routine can provide initial evidence from within the visual system for a causal relation in the environment that may then be integrated with signals from other domains (e.g. auditory signals). Determining exactly how the perception of causality relates to mechanisms of causal inference and the neural implementation thereof is an exciting avenue for future research. Note, however, that perceived causality can be distinguished from judged causality: Even when participants are aware that a third variable (e.g. a color change) is the best predictor of the movement of the second disc in launching events, they still perceive the first disc as causing the movement of the second disc (*Schlottmann and Shanks, 1992*).

Neurophysiological studies support the view of distributed neural processing underlying sensory causal interactions with the visual system playing a major role. Imaging studies in particular revealed a network for the perception of causality that is also involved in action observation (*Blakemore et al., 2003*; *Fonlupt, 2003*; *Fugelsang et al., 2005*; *Roser et al., 2005*). The fact that visual adaptation of causality occurs in a retinotopic reference frame emphasizes the role of retinotopically organized areas within that network (e.g. V5 and the superior temporal sulcus). Interestingly, single cell recordings in area F5 of the primate brain revealed that motor areas contribute to the perception of causality (*Caggiano et al., 2016*; *Rolfs, 2016*), emphasizing the distributed nature of the computations underlying causal interactions. This finding also stresses that the detection, and the prediction, of causality is essential for processes outside purely sensory systems (e.g. for understanding other's actions, for navigating, and for avoiding collisions). The neurophysiology subserving in causal inference further extends the candidate cortical areas that might contribute to the detection of causal relations, emphasizing the role of the frontal cortex for the flexible integration of multisensory representations (*Cao et al., 2019*; *Coen et al., 2023*).

Visual adaptation of causality constitutes a powerful tool that allows us to reveal how specialized visual routines are tuned to the motion direction of a launching event. Visual adaptation in general is thought to be one of the signatures of a perceptual process (*Hafri and Firestone, 2021*) and, therefore, is informative for identifying the visual origin of a mechanism as opposed to, for instance, cognitive accounts (e.g. cognitive schema; see *Rips, 2011* for a review). The detection of causal interactions features many hallmarks of perception: An impression of causality emerges fast and automatically for briefly presented stimuli (*Michotte, 1963*) and causal events reach awareness faster than non-causal events (*Moors et al., 2017*). The implementation of visual routines for the perception of causality in separate channels of a basic visual feature such as motion direction fortifies the view that our understanding of causal interactions indeed starts as a low-level perceptual routine.

## Materials and methods

### Participants

In each of the three experiments, we tested eight human observers (**Experiment 1**: ages 21–32 y; five female; three male; eight right-handed; five right-eye dominant: **Experiment 2**: ages 21–30 y; five female; three male; seven right-handed; six right-eye dominant; **Experiment 3**: ages 23–30 y; six female; one non-binary; seven right-handed; six right-eye dominant) in five sessions (one training session without adaptation, four test sessions with adaptation). We determined the final sample size by computing 90% power contours as a function of sample size and trial

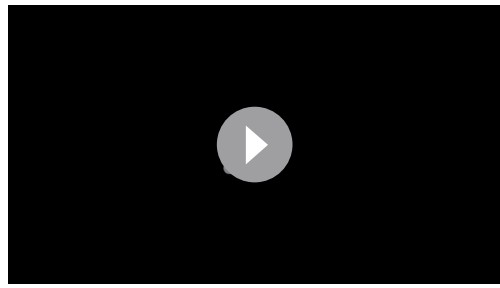

**Video 1.** Example of the launch stimulus. The stimulus is slowed down by a factor of 2.
https://elifesciences.org/articles/93454/figures#video1

number (*Baker et al., 2021*) and defining the additional criterion of a minimum of eight observers per experiment. Based on the data obtained in the session without adaptation (i.e. the first session), we determined whether participants do distinguish between passes and launches in the basic experiment by determining whether the proportion of reported passes increases with increasing disc overlap. We excluded observers who did not distinguish between passes and launches after the first session and replaced them by new observers (resulting in the replacement of one observer in each experiment). Data obtained in the first session (i.e. without adaptation) did not enter the final analyses. In **Experiment 1** and **2**, we paid observers 8€ per session as compensation for participation and a bonus of 4€ after successful completion of all sessions. In **Experiment 3**, we paid observers 10€ per session. In all experiments, we obtained observers' written informed consent before the first session. All observers had normal or corrected-to-normal vision and color vision as assessed using the color test *Tafeln zur Prüfung des Farbensinnes/Farbensehens* (*Kuchenbecker and Broschmann, 2016*). The study was approved by the ethics committee of the Psychology Department of the Humboldt-Universität zu Berlin and it followed the guidelines of the World Medical Association (2008). Declaration of Helsinki.

### Material

Observers sat in a sound-shielded, dimly lit room putting their head on a chin and forehead rest. We controlled for observers' eye position by tracking their dominant eye using an Eyelink 1000 Desktop Mount eye tracker (SR Research, Ottawa, ON, Canada) with a sampling rate of 1000 Hz. We displayed visual stimuli on a video-projection screen (Celexon HomeCinema, Tharston, Norwich, UK) using a PROPixx DLP projector (VPixx Technologies Inc, Saint Bruno, QC, Canada) at a spatial resolution of 1920×1080 pixels and a refresh rate of 120 Hz. The screen was mounted on a wall at 180 cm away from the observer. The experiment was running on a DELL Precision T7810 (Debian GNU Linux 8) and implemented in Matlab 2016b (Mathworks, Natick, MA, USA) using the Psychophysics toolbox 3 (*Brainard, 1997*; *Kleiner et al., 2007*; *Pelli, 1997*) for stimulus presentation and the Eyelink toolbox (*Cornelissen et al., 2002*) for control of the eye tracker. Behavioral responses were collected by pressing one of the two possible keys on a standard keyboard.

### General procedure

In our experiments, we asked observers to report whether they perceived a launch or a pass in test events composed of two discs. In the test events, a peripheral disc approached a stationary second disc and stopped at varying disc overlaps across trials (*Figure 1c*). Immediately after that, the second disc started moving in the same direction and with the same speed as the first disc. Before the first trial, observers read the instruction for the experiment that was displayed on the screen. As part of the instructions, we presented short demo trials of test events with 0% overlap that are

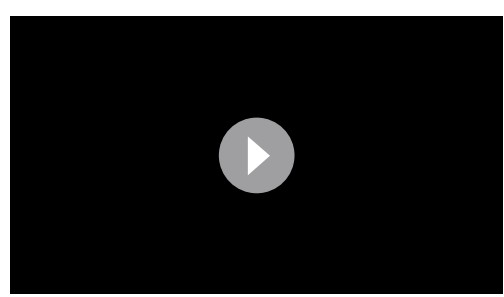

**Video 2.** Example of the pass stimulus. The stimulus is slowed down by a factor of 2.
https://elifesciences.org/articles/93454/figures#video2

typically perceived as launches (see *Video 1*), and test events with full disc overlap, that are typically perceived as a pass (see *Video 2*). Observers had the opportunity to inspect these two events as often as they wanted. Following the instructions, observers ran a short training session of 14 trials with varying disc overlaps ranging from 0–100% overlap.

At the beginning of a trial, we asked observers to fixate a gray fixation point (diameter of 1.5 dva) in the center of the screen on a black background and the trial only started after observers successfully fixated the fixation symbol for at least 200ms. We presented the test events at 3 dva below the fixation symbol. In these test events,

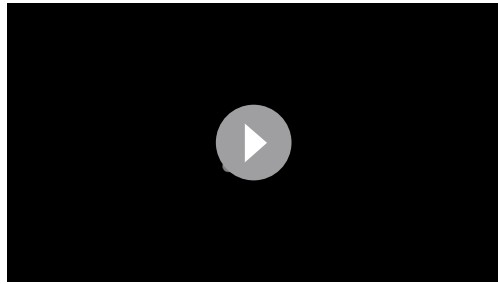

**Video 3.** Example of the slip stimulus. The stimulus is slowed down by a factor of 2.
https://elifesciences.org/articles/93454/figures#video3

the first moving disc (gray; diameter of 1.5 dva) located either left or right from the vertical meridian started moving towards a stationary disc (gray; diameter of 1.5 dva) located 3 dva below the fixation point. The first disc stopped moving at one of seven possible distances away from the stationary disc, resulting in seven different disc overlaps ranging from 0 to 100%. The entire duration of the test event was 175ms. At the end of the trial, observers reported whether they perceived a launch or a pass in the test event by pressing either the arrow up key for launches or arrow down key for passes. In our study, we assessed causal perception by asking observers to report whether they observed a launch or a pass in events of varying ambiguity. This method assumes that launches and passes can be mapped onto a dimension that ranges from causal to non-causal impressions. It has been questioned whether pass events are a natural representative of non-causal events: Observers often report high impressions of causality upon first exposure to pass events, which then decrease after seeing a canonical launch (*Bechlivanidis et al., 2019*). In our study, therefore, participants completed a separate session that included canonical launches before starting the main experiment.

In addition, the first session served the purpose to determine whether observers perceive launches and passes as a function of disc overlap in our behavioral task. To this end, observers completed 10 blocks with the direction of the test event (2 conditions: left vs. right) as well as the amount of disc overlap (7 conditions: ranging from 0–100% overlap) presented in a randomized order in a block. Each combination of these manipulations was repeated two times in a block, resulting in 28 trials in each block and a total of 280 trials in the first session. In sessions 2–5 we presented additional adaptors that varied between the experiments.

In all experiments, we tracked observers' dominant eye to ensure proper fixation behavior during presentation of the test events and presentation of the adaptors. More specifically, we tracked the dominant eye's current position at a sampling rate of 1000 Hz and determined online the eyes' distance to the screen center. We aborted a trial, whenever the distance between eye position and screen center exceeded 2 dva. Observers repeated these trials at the end of a block in randomized order. During presentation of the adaptors, we presented a short message (at the fixation point) asking observers to please fixate in the center of the screen once observers gaze exceeded 2 dva away from the screen center.

## Adaptation in Experiment 1

In **Experiment 1**, each observer ran in 15 blocks in each of sessions 2–5. The first 5 blocks in a session were again without adaptation to measure an observer's perception of causality before adaptation. As in the first session, we presented test events in two possible horizontal directions and varied the disc overlap resulting in 28 trials in each block. In blocks 6–15, we presented an adaptor before the first trial of a block. The adaptor was chosen from one of four possible adaptors. In two of the sessions, we used launches in a particular direction as the adaptor (the adaptor's direction was fixed in a session). Before the first trial of a block, we presented 320 launching events at the same location as the test events described above. The exact direction of these launching events was randomly chosen from a narrow uniform distribution around the direction on the horizontal meridian (±30 degrees). The long adaptation phase was complemented by a top-up adaptation of 16 launching events in the same

direction as the adaptor (again drawn from the same narrow uniform distribution around that main direction) before each trial to maintain an effective adaptation across the entire block.

In two additional sessions, we presented slip events as adaptors to control that the adaptation was specific for the impression of causality in the launching events (see *Video 3*). Slip events are designed to match the launching events in as many physical properties as possible while producing a very different, non-causal phenomenology. In slip events, the first peripheral disc also moves towards a stationary disc. In contrast to launching events, however, the first disc passes the stationary disc and stops only when it is adjacent to the opposite edge of the stationary disc. While slip events do not elicit a causal impression, they have the same number of objects and motion onsets, the same motion direction and speed, as well as the same spatial area of the event as launches. As for the launch adaptor, we displayed slip adaptors in a narrow uniform range of directions around one of the two possible horizontal directions (from left-to-right or from right-to-left being fixed in a session). This experimental design resulted in four sessions and the order of the type of adaptor (launches vs. slip adaptors; movement direction of the adaptor) was randomly determined for each participant. Overall, each observer ran in a total of 1,680 trials in **Experiment 1**.

## Adaptation variations in Experiment 2

In **Experiment 2**, we used a very similar design as in **Experiment 1** with one notable difference. Adaptors and test events varied in motion speed either being the same as in **Experiment 1** or half that speed. Correspondingly, in slow events, the test event duration was twice as long (i.e. event duration of 350 ms) as for test events displayed at fast motion speed (i.e. event duration of 175 ms). We presented all test events (and adaptors) always in the same direction (from left to right). Similarly, we also displayed slip events as adaptors in slow and fast motion speeds. In sessions with adaptation, the first five blocks were without adaptation while blocks 6–15 were with adaptors, each block consisting of 28 trials. Each observer ran in a total of 1680 trials in **Experiment 2**.

## Adaptation variations in Experiment 3

In **Experiment 3**, we determined the transfer of adaptation for feature conjunctions (motion direction x color). The two discs in the test events and during adaptation had different colors (i.e. red and green) and could move in both horizontal directions. The red and green discs were not matched for luminance. Measurements obtained after the experiments yielded a luminance of 21 $cd/m^2$ for the green disc and 6 $cd/m^2$ for the red disc. In contrast to **Experiments 1 and 2**, slip events were not used as adaptors. This design resulted in four different adaptors that were presented in separate sessions. Again, the first five blocks of a session were without adaptation while blocks 6–15 were with adaptors, each block consisting of 28 trials. Each observer ran in a total of 1,680 trials in **Experiment 3**.

## Data analysis

For the statistical analyses and estimation of the psychometric functions, we used the *quickpsy* package (*Linares and López-Moliner, 2016*) and the R environment (*Pelli, 1997*). We related disc overlap to the proportion of reported launches using logistic functions with four parameters for the intercept, slope, as well as upper and lower asymptotes. We fitted these functions separately for each observer and condition and obtained the points of subjective equality (PSE; the amount of overlap between the two discs in the test events that result in the same proportion of reported launches and passes). For model fitting, we constrained the range of possible estimates for each parameter of the logistic model. The lower asymptote for the proportion of reported launches was constrained to be in the range of 0–0.75, and the upper asymptote in the range of 0.25–1. The intercept of the logistic model was constrained to be in the range 1–15, and the slope was constrained to be in the range –20 to –1.

For inferential statistics, we analyzed PSEs using repeated-measures analyses of variance (rmANOVA). Error bars indicate ±1 within-subject standard error of the mean (SEM; *Baguley, 2012*; *Morey, 2008*). A significant interaction in the rmANOVA was complemented by running post-hoc paired t-tests. Please note, one observer in **Experiment 3** showed such a strong negative aftereffect when test event and adaptor were identical, such that the upper asymptote was below a proportion of 0.5 (i.e. the point when launches and passes are equally often reported). Instead of a non-identifiable PSE for that observer in that particular condition, we computed inflection points of the psychometric functions in all conditions for that observer which then entered the statistical analyses. Neither of the

studies reported in this article was preregistered. The data and analysis code has been deposited in the Open Science Framework and is publicly available at https://osf.io/x947m/.

## Acknowledgements

This research was supported by a DFG research grant to SO (OH 274/4–1) as well as funding from the Heisenberg Programme of the DFG to SO (OH 274/5–1) and MR (grants RO3579/8-1 and RO3579/12-1). We acknowledge support by the Open Access Publication Fund of Humboldt-Universität zu Berlin. The authors declare no competing financial interests.

## Additional information

### Funding

| Funder | Grant reference number | Author |
|---|---|---|
| Deutsche Forschungsgemeinschaft | OH 274/4-1 | Sven Ohl |
| Deutsche Forschungsgemeinschaft | OH 274/5-1 | Sven Ohl |
| Deutsche Forschungsgemeinschaft | RO3579/8-1 | Martin Rolfs |
| Deutsche Forschungsgemeinschaft | RO3579/12-1 | Martin Rolfs |

The funders had no role in study design, data collection and interpretation, or the decision to submit the work for publication.

### Author contributions

Sven Ohl, Conceptualization, Resources, Data curation, Software, Formal analysis, Funding acquisition, Validation, Investigation, Visualization, Methodology, Writing – original draft, Project administration, Writing – review and editing; Martin Rolfs, Conceptualization, Resources, Software, Methodology, Writing – review and editing

### Author ORCIDs

Sven Ohl https://orcid.org/0000-0002-0292-2151
Martin Rolfs https://orcid.org/0000-0002-8214-8556

### Ethics

In all experiments, we obtained observers' written informed consent before the first session. The study was approved by the ethics committee of the Psychology Department of the Humboldt-Universität zu Berlin and it followed the guidelines of the Declaration of Helsinki (2008).

Reviewer #1 (Public review): https://doi.org/10.7554/eLife.93454.3.sa1
Reviewer #2 (Public review): https://doi.org/10.7554/eLife.93454.3.sa2
Reviewer #3 (Public review): https://doi.org/10.7554/eLife.93454.3.sa3
Author response https://doi.org/10.7554/eLife.93454.3.sa4

## Additional files

### Supplementary files
MDAR checklist

### Data availability
The data and analysis code has been deposited at the Open Science Framework and is publicly available at https://osf.io/x947m/.

The following dataset was generated:

| Author(s) | Year | Dataset title | Dataset URL | Database and Identifier |
|-----------|------|---------------|-------------|------------------------|
| Ohl S, Rolfs M | 2025 | Visual routines for detecting causal interactions are tuned to motion direction | https://osf.io/x947m/ | Open Science Framework, x947m |

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
