## [Editor Report · eLife Assessment]

This study provides a **valuable** contribution to our understanding of causal inference in visual perception. The evidence provided through multiple well-designed psychophysical experiments is **convincing**. The current study targets very specific visual features of launch events, future work will be able to build on this to study the implementation of causal inference in general.

---

## [Referee Report · Reviewer #1 (Public review)]

Summary:

The authors investigated causal inference in the visual domain through a set of carefully designed experiments, and sound statistical analysis. They suggest the early visual system has a crucial contribution to computations supporting causal inference.

Strengths:

(1) I believe the authors target an important problem (causal inference) with carefully chosen tools and methods. Their analysis rightly implies the specialization of visual routines for causal inference and the crucial contribution of early visual systems to perform this computation. I believe this is a novel contribution and their data and analysis are in the right direction.

(2) Authors sufficiently discuss the alternative perspective to causal inference.

(3) The authors also expand the discussions beyond pure psychophysics and also include neural aspects.

Weaknesses:

I would not call them weaknesses, perhaps a different perspective:

(1) Authors arguing pro a mere bottom-up contribution of early sensory areas for causal inference. Certainly, as the authors suggested, early sensory areas have a crucial contribution, and the authors expand it to other possibilities in their discussion (but more for more complex scenario). It would say, even in simple cases, we can still consider the effect of top down processes. This particularly makes sense in light of recent studies. These studies progressively suggest perception as an active process that also weighs in strongly, the top-down cognitive contributions. For instance, the most simple cases of perception have been conceptualized along this line (Martin, Solms, and Sterzer 2021) and even some visual illusions (Safavi and Dayan 2022), and other extensions (Kay et al. 2023). Thus, I believe it would be helpful to extend the discussion on the top-down and cognitive contributions of causal inference (of course that can also be hinted at, based on recent developments). Even adaptation, which is central in this study, can be influenced by top-down factors (Keller et al. 2017).

Lastly, I hope the authors find this review helpful. I generally want to try to end all of my reviews with areas of the paper I liked because I think this should be part of the feedback. Certainly, there were many in this manuscript as well (clever questions, experimental design and statistical analysis) that I had to highlight further. I congratulate the authors again on their manuscript and hope they will find it helpful.

Bibliography

Aller, Mate, and Uta Noppeney. 2018. "To Integrate or Not to Integrate: Temporal Dynamics of Bayesian Causal Inference." Biorxiv, December, 504118. .

Cao, Yinan, Christopher Summerfield, Hame Park, Bruno Lucio Giordano, and Christoph Kayser. 2019. "Causal Inference in the Multisensory Brain." Neuron 102 (5): 1076-87.e8. .

Coen, Philip, Timothy P. H. Sit, Miles J. Wells, Matteo Carandini, and Kenneth D. Harris. 2021. "The Role of Frontal Cortex in Multisensory Decisions." Biorxiv, April. Cold Spring Harbor Laboratory, 2021.04.26.441250. .

Kay, Kendrick, Kathryn Bonnen, Rachel N. Denison, Mike J. Arcaro, and David L. Barack. 2023. "Tasks and Their Role in Visual Neuroscience." Neuron 111 (11). Elsevier: 1697-1713. .

Keller, Andreas J, Rachael Houlton, Björn M Kampa, Nicholas A Lesica, Thomas D Mrsic-Flogel, Georg B Keller, and Fritjof Helmchen. 2017. "Stimulus Relevance Modulates Contrast Adaptation in Visual Cortex." Elife 6. eLife Sciences Publications, Ltd: e21589.

Kording, K. P., U. Beierholm, W. J. Ma, S. Quartz, J. B. Tenenbaum, and L. Shams. 2007. "Causal Inference in Multisensory Perception." PloS One 2: e943. .

Martin, Joshua M., Mark Solms, and Philipp Sterzer. 2021. "Useful Misrepresentation: Perception as Embodied Proactive Inference." Trends Neurosci. 44 (8): 619-28. .

Safavi, Shervin, and Peter Dayan. 2022. "Multistability, Perceptual Value, and Internal Foraging." Neuron, August. .

Shams, L. 2012. "Early Integration and Bayesian Causal Inference in Multisensory Perception." In The Neural Bases of Multisensory Processes, edited by M. M. Murray and M. T. Wallace. Frontiers in Neuroscience. Boca Raton (FL).

Shams, Ladan, and Ulrik Beierholm. 2022. "Bayesian Causal Inference: A Unifying Neuroscience Theory." Neuroscience & Biobehavioral Reviews 137 (June): 104619.

---

## [Referee Report · Reviewer #2 (Public review)]

This paper seeks to determine whether the human visual system's sensitivity to causal interactions is tuned to specific parameters of a causal launching event, using visual adaptation methods. The three parameters the author investigates in this paper are the direction of motion in the event, the speed of the objects in the event, and surface features or identity of the objects in the event (in particular, having two objects of different color).

The key method, visual adaptation to causal launching, has now been demonstrated by at least three separate groups and seems to be a robust phenomenon. Adaptation is a strong indicator of a visual process that is tuned to a specific feature of the environment, in this case launching interactions. Whereas other studies have focused on retinotopically-specific adaptation (i.e., whether the adaptation effect is restricted to the same test location on the retina as the adaptation stream was presented to), this one focuses on feature-specificity.

The first experiment replicates the adaptation effect for launching events as well as the lack of adaptation event for a minimally different non-causal 'slip' event. However, it also finds that the adaptation effect does not work for launching events that do not have a direction of motion more than 30 degrees from the direction of the test event. The interpretation is that the system that is being adapted is sensitive to the direction of this event, which is an interesting and somewhat puzzling result given the methods used in previous studies, which have used random directions of motion for both adaptation and test events.

The obvious interpretation would be that past studies have simply adapted to launching in every direction, but that in itself says something about the nature of this direction-specificity: it is not working through opposed detectors. For example, in something like the waterfall illusion adaptation effect, where extended exposure to downward motion leads to illusory upward motion on neutral-motion stimuli, the effect simply doesn't work if motion in two opposed directions are shown (i.e., you don't see illusory motion in both directions, you just see nothing). The fact that adaptation to launching in multiple directions doesn't seem to cancel out the adaptation effect in past work raises interesting questions about how directionality is being coded in the underlying process. In addition, one limitation of the current method is that it's not clear whether the motion-direction-specificity is also itself retinotopically-specific, that is, if one retinotopic location were adapted to launching in one direction and a different retinotopic location adapted to launching in the opposite direction, would each test location show the adaptation effect only for events in the direction presented at that location?

The second experiment tests whether the adaptation effect is similarly sensitive to differences in speed. The short answer is no; adaptation events at one speed affect test events at another. Furthermore, this is not surprising given that Kominsky & Scholl (2020) showed adaptation transfer between events with differences in speeds of the individual objects in the event (whereas all events in this experiment used symmetrical speeds). This experiment is still novel and it establishes that the speed-insensitivity of these adaptation effects is fairly general, but I would certainly have been surprised if it had turned out any other way.

The third experiment tests color (as a marker of object identity), and pits it against motion direction. The results demonstrate that adaptation to red-launching-green generates an adaptation effect for green-launching-red, provided they are moving in roughly the same direction, which provides a nice internal replication of Experiment 1 in addition to showing that the adaptation effect is not sensitive to object identity. This result forms an interesting contrast with the infant causal perception literature. Multiple papers (starting with Leslie & Keeble, 1987) have found that 6-8-month-old infants are sensitive to reversals in causal roles exactly like the ones used in this experiment. The success of adaptation transfer suggests, very clearly, that this sensitivity is not based only on perceptual processing, or at least not on the same processing that we access with this adaptation procedure. It implies that infants may be going beyond the underlying perceptual processes and inferring genuine causal content. This is also not the first time the adaptation paradigm has diverged from infant findings: Kominsky & Scholl (2020) found a divergence with the object speed differences as well, as infants categorize these events based on whether the speed ratio (agent:patient) is physically plausible (Kominsky et al., 2017), while the adaptation effect transfers from physically implausible events to physically plausible ones. This only goes to show that these adaptation effects don't exhaustively capture the mechanisms of early-emerging causal event representation.

One overarching point about the analyses to take into consideration: The authors use a Bayesian psychometric curve-fitting approach to estimate a point of subjective equality (PSE) in different blocks for each individual participant based on a model with strong priors about the shape of the function and its asymptotic endpoints, and this PSE is the primary DV across all of the studies. As discussed in Kominsky & Scholl (2020), this approach has certain limitations, notably that it can generate nonsensical PSEs when confronted with relatively extreme response patterns. The authors mentioned that this happened once in Experiment 3, and that participant had to be replaced. An alternate approach is simply to measure the proportion of 'pass' reports overall to determine if there is an adaptation effect. The results here do not change based on which analytical strategy is used, which ultimately just goes to show that the effects are very robust.

In general, this paper adds further evidence for something like a 'launching' detector in the visual system, but beyond that it specifies some interesting questions for future work about how exactly such a detector might function.

Kominsky, J. F., & Scholl, B. J. (2020). Retinotopic adaptation reveals distinct categories of causal perception. Cognition, 203, 104339. https://doi.org/10.1016/j.cognition.2020.104339

Kominsky, J. F., Strickland, B., Wertz, A. E., Elsner, C., Wynn, K., & Keil, F. C. (2017). Categories and Constraints in Causal Perception. Psychological Science, 28(11), 1649-1662. https://doi.org/10.1177/0956797617719930

Leslie, A. M., & Keeble, S. (1987). Do six-month-old infants perceive causality? Cognition, 25(3), 265-288. https://doi.org/10.1016/S0010-0277(87)80006-9

---

## [Referee Report · Reviewer #3 (Public review)]

Summary:

This paper presents evidence from three behavioral experiments that causal impressions of "launching events", in which one object is perceived to cause another object to move, depend on motion direction-selective processing. Specifically, the work uses an adaptation paradigm (Rolfs et al., 2013), presenting repetitive patterns of events matching certain features to a single retinal location, then measuring subsequent perceptual reports of a test display in which the degree of overlap between two discs was varied, and participants could respond "launch" or "pass". The three experiments report results of adapting to motion direction, motion speed and "object identity", and examine how the psychometric curves for causal reports shift in these conditions depending on the similarity of adapter and test. While causality reports in the test display were selective for motion direction (Experiment 1), they were not selective for adapter-test speed differences (Experiment 2) nor for changes in object identity induced via color swap (Experiment 3). These results support the notion of a biological implementation of causality perception in the visual system, possibly even independently of computations of object identity.

Strengths:

The setup of the research question and hypotheses are exceptional. The authors thoroughly discuss relevant literature to clearly link their launch/pass paradigm to impressions of causality, strengthening their hypothesis and conclusions. The experiments are carefully performed (appropriate equipment, careful control of eye movements). The slip adaptor is a really nice control condition and effectively mitigates the need to control for motion direction with a drifting grating or similar. Participants were measured with sufficient precision, and a power curve analysis was conducted to determine the sample size. Data analysis and statistical quantification is appropriate. Data and analysis code will be shared on publication, in keeping with open science principles. The paper is concise and well written.

Weaknesses:

I would like to emphasise that in the employed paradigm and previously conducted similar study, the only report options are "launch" or "pass". As pointed out by the authors' reply, the adaptation to launches seems to be a highly specific process and likely is a consequence of the causal interaction between the objects. I would nonetheless be interested to see which of the stimulus features driving the adaptation effect observed here are relevant/irrelevant to subjective causal impressions in an experiment.

References:

Rolfs, M., Dambacher, M., & Cavanagh, P. (2013). Visual Adaptation of the Perception of Causality. Current Biology, 23(3), 250-254. https://doi.org/10.1016/j.cub.2012.12.017

---

## [Author Response]

The following is the authors’ response to the original reviews.

**Reviewer #1 (Public Review**):Summary:The authors investigated causal inference in the visual domain through a set of carefully designed experiments, and sound statistical analysis. They suggest the early visual system has a crucial contribution to computations supporting causal inference.Strengths:I believe the authors target an important problem (causal inference) with carefully chosen tools and methods. Their analysis rightly implies the specialization of visual routines for causal inference and the crucial contribution of early visual systems to perform this computation. I believe this is a novel contribution and their data and analysis are in the right direction.Weaknesses:In my humble opinion, a few aspects deserve more attention:(1) Causal inference (or causal detection) in the brain should be quite fundamental and quite important for human cognition/perception. Thus, the underlying computation and neural substrate might not be limited to the visual system (I don't mean the authors did claim that). In fact, to the best of my knowledge, multisensory integration is one of the best-studied perceptual phenomena that has been conceptualized as a causal inference problem.Assuming the causal inference in those studies (Shams 2012; Shams and Beierholm 2022; Kording et al. 2007; Aller and Noppeney 2018; Cao et al. 2019) (and many more e.g., by Shams and colleagues), and the current study might share some attributes, one expects some findings in those domains are transferable (at least to some degree) here as well. Most importantly, underlying neural correlates that have been suggested based on animal studies and invasive recording that has been already studied, might be relevant here as well.Perhaps the most relevant one is the recent work from the Harris group on mice (Coen et al. 2021). I should emphasize, that I don't claim they are necessarily relevant, but they can be relevant given their common roots in the problem of causal inference in the brain. This is a critical topic that the authors may want to discuss in their manuscript.

We thank the reviewer. We addressed this point of the public review in our reply to the reviewer’s suggestions (and add it here again for convenience). The literature on the role of occipital, parietal and frontal brain areas in causal inference is also addressed in the response to point 3 of the public review.

“We used visual adaptation to carve out a bottom-up visual routine for detecting causal interactions in form of launching events. However, we know that more complex behaviors of perceiving causal relations can result from integrating information across space (e.g., in causal capture; Scholl & Nakayama, 2002), across time (postdictive influence; Choi & Scholl, 2006), and across sensory modalities (Sekuler, Sekuler, & Lau, 1997). Bayesian causal inference has been particularly successful as a normative framework to account for multisensory integration (Körding et al., 2007; Shams & Beierholm, 2022). In that framework, the evidence for a common-cause hypothesis is competing with the evidence for an independent-causes hypothesis (Shams & Beierholm, 2022). The task in our experiments could be similarly formulated as two competing hypotheses for the second disc’s movement (i.e., the movement was caused by the first disc vs. the movement occurred autonomously). This framework also emphasizes the distributed nature of the neural implementation for solving such inferences, showing the contributions of parietal and frontal areas in addition to sensory processing (for review see Shams & Beierholm, 2022). Moreover, even visual adaptation to contrast in mouse primary visual cortex is influenced by top-down factors such as behavioral relevance— suggesting a complex implementation of the observed adaptation results (Keller et al. 2017). The present experiments, however, presented purely visual events that do not require an integration across processing domains. Thus, the outcome of our suggested visual routine can provide initial evidence from within the visual system for a causal relation in the environment that may then be integrated with signals from other domains (e.g., auditory signals). Determining exactly how the perception of causality relates to mechanisms of causal inference and the neural implementation thereof is an exciting avenue for future research. Note, however, that perceived causality can be distinguished from judged causality: Even when participants are aware that a third variable (e.g., a color change) is the best predictor of the movement of the second disc in launching events, they still perceive the first disc as causing the movement of the second disc (Schlottmann & Shanks, 1992).”

(2) If I understood correctly, the authors are arguing pro a mere bottom-up contribution of early sensory areas for causal inference (for instance, when they wrote "the specialization of visual routines for the perception of causality at the level of individual motion directions raises the possibility that this function is located surprisingly early in the visual system *as opposed to a higher-level visual computation*."). Certainly, as the authors suggested, early sensory areas have a crucial contribution, however, it may not be limited to that. Recent studies progressively suggest perception as an active process that also weighs in strongly, the topdown cognitive contributions. For instance, the most simple cases of perception have been conceptualized along this line (Martin, Solms, and Sterzer 2021) and even some visual illusion (Safavi and Dayan 2022), and other extensions (Kay et al. 2023). Thus, I believe it would be helpful to extend the discussion on the top-down and cognitive contributions of causal inference (of course that can also be hinted at, based on recent developments). Even adaptation, which is central in this study can be influenced by top-down factors (Keller et al. 2017). I believe, based on other work of Rolfs and colleagues, this is also aligned with their overall perspective on vision.

Indeed, we assessed bottom-up contributions to the perception of a causal relation. We agree with the reviewer that in more complex situations, for instance, in the presence of contextual influences or additional auditory signals, the perception of a causal relation may not be limited to bottom-up vision. While we had acknowledged this in the original manuscript (see excerpts below), we now make it even more explicit:

“[…] we know that more complex behaviors of perceiving causal relations can result from integrating information across space (e.g., in causal capture; Scholl & Nakayama, 2002), across time (postdictive influence; Choi & Scholl, 2006), and across sensory modalities (Sekuler, Sekuler, & Lau, 1997).”

“[…] Neurophysiological studies support the view of distributed neural processing underlying sensory causal interactions with the visual system playing a major role.”

“[…] Interestingly, single cell recordings in area F5 of the primate brain revealed that motor areas are contributing to the perception of causality (Caggiano et al., 2016; Rolfs, 2016), emphasizing the distributed nature of the computations underlying causal interactions. This finding also stresses that the detection, and the prediction, of causality is essential for processes outside sensory systems (e.g., for understanding other’s actions, for navigating, and for avoiding collisions). The neurophysiology subserving causal inference further extend the candidate cortical areas that might contibute to the detection of causal relations, emphasizing the role of the frontal cortex for the flexible integration of multisensory representations (Cao et al., 2019; Coen et al., 2023).”

However, there is also ample evidence that the perception of a simple causal relation—as we studied it in our experiments—escapes top-down cognitive influences. The perception of causality in launching events is described as automatic and irresistible, meaning that participants have the spontaneous impression of a causal relation, and participants typically do not voluntarily switch between a causal and a noncausal percept. This irresistibility has led several authors to discuss a modular organization underlying the detection of such events (Michotte, 1963; Scholl & Tremoulet, 2000). This view is further supported by a study that experimentally manipulated the contingencies between the movement of the two discs (Schlottmann & Shanks, 1992). In one condition the authors created a launching event where the second disc’s movement was perfectly correlated with a color change, but only sometimes coincided with the first disc’s movement offset. Nevertheless, participants reported seeing that the first disc caused the movement of second disc (regardless of the stronger statistical relationship with the color change). However, when asked to make conscious causal judgments, participants were aware of the color change as the true cause of the second disc’s motion—therefore recognizing its more reliable correlation. This study strongly suggests that perceived and judged causality (i.e., cognitive causal inference) can be dissociated (Schlottmann & Shanks, 1992). We have added this reference in the revised manuscript. Overall, we argue that our study focused on a visual routine that could be implemented in a simple bottom-up fashion, but we acknowledge throughout the manuscript, that in a more complex situation (e.g., integrating information from other sensory domains) the implementation could be realized in a more distributed fashion including top-down influences as in multisensory integration. However, it is important to stress that these potential top-down influences would be automatic and should not be confused with voluntary cognitive influences.

“Note, however, that perceived causality can be distinguished from judged causality (Schlottmann & Shanks, 1992). Even when participants are aware that a third variable (e.g., a color change) is the best predictor of the movement of the second disc in launching events, they still perceive the first disc as causing the movement of the second disc (Schlottmann & Shanks, 1992).”

(3) The authors rightly implicate the neural substrate of causal inference in the early sensory system. Given their study is pure psychophysics, a more elaborate discussion based on other studies that used brain measurements is needed (in my opinion) to put into perspective this conclusion. In particular, as I mentioned in the first point, the authors mainly discuss the potential neural substrate of early vision, however much has been done about the role of higher-tier cortical areas in causal inference e.g., see (Cao et al. 2019; Coen et al. 2021).

In the revised manuscript, we addressed the limitations of a purely psychophysical approach and acknowledged alternative implementations in the Discussion section.

“Note that, while the present findings demonstrate direction-selectivity, it remains unclear where exactly that visual routine is located. As pointed out, it is also possible that the visual routine is located higher up in the visual system (or distributed across multiple levels) and is only using a directional-selective population response as input.”

Moreover, we cite also the two suggested papers when referring to the role of cortical areas in causal inference (Cao et al, 2019; Coen et al., 2023):

“Neurophysiological studies support the view of distributed neural processing underlying sensory causal interactions with the visual system playing a major role. Imaging studies in particular revealed a network for the perception of causality that is also involved in action observation (Blakemore et al., 2003; Fonlupt, 2003; Fugelsang et al., 2005; Roser et al., 2005). The fact that visual adaptation of causality occurs in a retinotopic reference frame emphazises the role of retinotopically organized areas within that network (e.g., V5 and the superior temporal sulcus). Interestingly, single cell recordings in area F5 of the primate brain revealed that motor areas are contributing to the perception of causality (Caggiano et al., 2016; Rolfs, 2016), emphasizing the distributed nature of the computations underlying causal interactions, and also stressing that the detection, and the prediction, of causality is essential for processes outside purely sensory systems (e.g., for understanding other’s actions, for navigating, and for avoiding collisions). The neurophysiological underpinnings in causal inference further extend the candidate cortical areas that might contibute to the detection of causal relations, emphasizing the role of the frontal cortex for the flexible integration of multisensory representations (Cao et al., 2019; Coen et al., 2023).”

There were many areas in this manuscript that I liked: clever questions, experimental design, and statistical analysis.

Thank you so much.

**Reviewer #1 (Recommendations for the authors):**
I congratulate the authors again on their manuscript and hope they will find my review helpful. Most of my notes are suggestions to the authors, and I hope will help them to improve the manuscript. None are intended to devalue their (interesting) work.

We would like to thank the reviewer for their thoughtful and encouraging comments.

In the following, I use pX-lY template to refer to a particular page number, say page number X (pX), and line number, say line number Y (lY).Major concerns and suggestions- I would suggest simplifying the abstract and significance statement or putting more background in it. It's hard (at least for me) to understand if one is not familiar with the task used in this study.

We followed the reviewer’s suggestion and added more background in the beginning of the abstract.

We made the following changes:

“Detecting causal relations structures our perception of events in the world. Here, we determined for visual interactions whether generalized (i.e., feature-invariant) or specialized (i.e., feature-selective) visual routines underlie the perception of causality. To this end, we applied a visual adaptation protocol to assess the adaptability of specific features in classical launching events of simple geometric shapes. We asked observers to report whether they observed a launch or a pass in ambiguous test events (i.e., the overlap between two discs varied from trial to trial). After prolonged exposure to causal launch events (the adaptor) defined by a particular set of features (i.e., a particular motion direction, motion speed, or feature conjunction), observers were less likely to see causal launches in subsequent ambiguous test events than before adaptation. Crucially, adaptation was contingent on the causal impression in launches as demonstrated by a lack of adaptation in non-causal control events. We assessed whether this negative aftereffect transfers to test events with a new set of feature values that were not presented during adaptation. Processing in specialized (as opposed to generalized) visual routines predicts that the transfer of visual adaptation depends on the feature-similarity of the adaptor and the test event. We show that negative aftereffects do not transfer to unadapted launch directions but do transfer to launch events of different speed. Finally, we used colored discs to assign distinct feature-based identities to the launching and the launched stimulus. We found that the adaptation transferred across colors if the test event had the same motion direction as the adaptor. In summary, visual adaptation allowed us to carve out a visual feature space underlying the perception of causality and revealed specialized visual routines that are tuned to a launch’s motion direction.”

- The authors highlight the importance of studying causal inference and understanding the underlying mechanisms by probing adaptation, however, their introduction justifying that is, in my humble opinion, quite short. Perhaps in the cited paper, this is discussed extensively, but I'd suggest providing some elaboration in the manuscript. Otherwise, the study would be very specific to certain visual phenomena, rather than general mechanisms.

We have carefully considered the reviewer’s set of comments and concerns (e.g., the role of top-down influences, the contributions of the frontal cortex, and illustration of the computational level). They all appear to share the theme that the reviewer looks at our study from the perspective of Bayesian inference. We conducted the current study in the tradition of classical phenomena in the field of the perception of causality (in the tradition of Michotte, 1963 and as reviewed in Scholl & Tremoulet, 2000) which aims to uncover the relevant visual parameters and rules for detecting causal relations in the visual domain. Indeed, we think that a causal inference perspective promises a lot of new insights into the mechanisms underlying the classical phenomena described for the perception of causality. In the revised manuscript, we discuss therefore causal inference and how it relates to the current study. We now emphasize that in our study, (a) we used visual adaptation to reveal the bottom-up processes that allow for the detection of a causal interaction in the visual domain, (b) that the perception of causality also integrates signals from other domains (which we do not study here), and (c) that the neural substrates underlying the perception of causality might be best described by a distributed network. By discussing Bayesian causal inference, we point out promising avenues for future research that may bridge the fields of the perception of causality and Bayesian causal inference. However, we also emphasize that perceived causality and judged causality can be dissociated (Schlottmann & Shanks, 1992).

We added the following discussion:

“We used visual adaptation to carve out a bottom-up visual routine for detecting causal interactions in form of launching events. However, we know that more complex behaviors of perceiving causal relations can result from integrating information across space (e.g., in causal capture; Scholl & Nakayama, 2002), across time (postdictive influence; Choi & Scholl, 2006), and across sensory modalities (Sekuler, Sekuler, & Lau, 1997). Bayesian causal inference has been particularly successful as a normative framework to account for multisensory integration (Körding et al., 2007; Shams & Beierholm, 2022). In that framework, the evidence for a common-cause hypothesis is competing with the evidence for an independent-causes hypothesis (Shams & Beierholm, 2022). The task in our experiments could be similarly formulated as two competing hypotheses for the second disc’s movement (i.e., the movement was caused by the first disc vs. the second disc did not move). This framework also emphasizes the distributed nature of the neural implementation for solving such inferences, showing the contributions of parietal and frontal areas in addition to sensory processing (for review see Shams & Beierholm, 2022). Moreover, even visual adaptation to contrast in mouse primary visual cortex is influenced by top-down factors such as behavioral relevance— suggesting a complex implementation of the observed adaptation results (Keller et al. 2017). The present experiments, however, presented purely visual events that do not require an integration across processing domains. Thus, the outcome of our suggested visual routine can provide initial evidence from within the visual system for a causal relation in the environment that may then be integrated with signals from other domains (e.g., auditory signals). Determining exactly how the perception of causality relates to mechanisms of causal inference and the neural implementation thereof is an exciting avenue for future research. Note, however, that perceived causality can be distinguished from judged causality: Even when participants are aware that a third variable (e.g., a color change) is the best predictor of the movement of the second disc in launching events, they still perceive the first disc as causing the movement of the second disc (Schlottmann & Shanks, 1992).”

- I'd suggest, at the outset, already set the context, that your study of causal inference in the brain is specifically targeting the visual domain, if you like, in the discussion connect it better to general ideas about causal inference in the brain (like the works by Ladan Shams and colleagues).

We would like to thank the reviewer for this comment. We followed the reviewer’s suggestion and made clear from the beginning that this paper is about the detection of causal relations in the visual domain. In the revised manuscript we write:

“Here, we will study the mechanisms underlying the computations of causal interactions in the visual domain by capitalizing on visual adaptation of causality (Kominsky & Scholl, 2020; Rolfs et al., 2013). Adaptation is a powerful behavioral tool for discovering and dissecting a visual mechanism (Kohn, 2007; Webster, 2015) that provides an intriguing testing ground for the perceptual roots of causality.”

As described in our reply to the previous comment, we now also discussed the ideas about causal inference.

- To better illustrate the implication of your study on the computational level, I'd suggest putting it in the context of recent approaches to perception (point 2 of my public review). I think this is also aligned with the comment of Reviewer#3 on your line 32 (recommendation for authors).

In the revised manuscript, we now discuss the role of top-down influences in causal inference when addressing point 2 of the reviewer’s public review.

Minor concerns and suggestions- On p2-l3, I'd suggest providing a few examples for generalized and or specialized visual routines (given the importance of the abstract). I only got it halfway through the introduction.

We thank the reviewer for highlighting the need to better introduce the concept of a visual routine. We have chosen the term visual routine to emphasize that we locate the part of the mechanism that is affected by the adaptation in our experiments in the visual system. At the same time, the concept leaves space with respect to the extent to which the mechanism further involves mid- and higher-level processes. In the revised manuscript, we now refer to Ullman (1987) who introduced the concept of a visual routine—the idea of a modular operation that sequentially processes spatial and feature information. Moreover, we refer to the concept of attentional sprites (Cavanagh, Labianca, & Thornton, 2001)—attention-based visual routines that allow the visual system to semi-independently handle complex visual tasks (e.g., identifying biological motion).

We add the following footnote to the introduction:

“We use the term visual routine here to highlight that our adaptation experiments can reveal a causality detection mechanism that resides in the visual system. At the same time, calling it a routine emphasizes similarities with a local, semi-independent operation (e.g., the recognition of familiar motion patterns; see also Ullman, 1987; Cavanagh, Labianca, & Thornton, 2001) that can engage mid- and higher-level processes (e.g., during causal capture, Scholl & Nakayama, 2002; or multisensory integration, Körding et al., 2007).”

In the abstract we now write:

“Here, we determined for visual interactions whether generalized (i.e., feature-invariant) or specialized (i.e., feature-selective) visual routines underlie the perception of causality.”

- On p4-l31, I'd suggest mentioning the Matlab version. I have experienced differences across different versions of Matlab (minor but still ...).

We added the Matlab Version.

- On p6-l46 OSF-link is missing (that contains data and code).

Thank you. We made the OSF repository public and added the link to the revised manuscript.

We added the following information to the revised manuscript.

“The data analysis code has been deposited at the Open Science Framework and is publicly available https://osf.io/x947m/.”

**Reviewer #2 (Public Review):**
This paper seeks to determine whether the human visual system's sensitivity to causal interactions is tuned to specific parameters of a causal launching event, using visual adaptation methods. The three parameters the authors investigate in this paper are the direction of motion in the event, the speed of the objects in the event, and the surface features or identity of the objects in the event (in particular, having two objects of different colors). The key method, visual adaptation to causal launching, has now been demonstrated by at least three separate groups and seems to be a robust phenomenon. Adaptation is a strong indicator of a visual process that is tuned to a specific feature of the environment, in this case launching interactions. Whereas other studies have focused on retinotopically specific adaptation (i.e., whether the adaptation effect is restricted to the same test location on the retina as the adaptation stream was presented to), this one focuses on feature specificity.The first experiment replicates the adaptation effect for launching events as well as the lack of adaptation event for a minimally different non-causal 'slip' event. However, it also finds that the adaptation effect does not work for launching events that do not have a direction of motion more than 30 degrees from the direction of the test event. The interpretation is that the system that is being adapted is sensitive to the direction of this event, which is an interesting and somewhat puzzling result given the methods used in previous studies, which have used random directions of motion for both adaptation and test events.The obvious interpretation would be that past studies have simply adapted to launching in every direction, but that in itself says something about the nature of this direction-specificity: it is not working through opposed detectors. For example, in something like the waterfall illusion adaptation effect, where extended exposure to downward motion leads to illusory upward motion on neutral-motion stimuli, the effect simply doesn't work if motion in two opposed directions is shown (i.e., you don't see illusory motion in both directions, you just see nothing). The fact that adaptation to launching in multiple directions doesn't seem to cancel out the adaptation effect in past work raises interesting questions about how directionality is being coded in the underlying process.

We would like to thank the reviewer for that thoughtful comment. We added the described implication to the manuscript:

“While the present study demonstrates direction-selectivity for the detection of launches, previous adaptation protocols demonstrated successful adaptation using adaptors with random motion direction (Rolfs et al., 2013; Kominsky & Scholl, 2020). These results therefore suggest independent direction-specific routines, in which adaptation to launches in one direction does not counteract an adaptation to launches in the opposite direction (as for example in opponent color coding).”

In addition, one limitation of the current method is that it's not clear whether the motion direction-specificity is also itself retinotopically-specific, that is, if one retinotopic location were adapted to launching in one direction and a different retinotopic location adapted to launching in the opposite direction, would each test location show the adaptation effect only for events in the direction presented at that location?

This is an interesting idea! Because previous adaptation studies consistently showed retinotopic adaptation of causality, we would not expect to find transfer of directional tuning for launches to other locations. We agree that the suggested experiment on testing the reference frame of directional specificity constitutes an interesting future test of our findings.

The second experiment tests whether the adaptation effect is similarly sensitive to differences in speed. The short answer is no; adaptation events at one speed affect test events at another. Furthermore, this is not surprising given that Kominsky & Scholl (2020) showed adaptation transfer between events with differences in speeds of the individual objects in the event (whereas all events in this experiment used symmetrical speeds). This experiment is still novel and it establishes that the speed-insensitivity of these adaptation effects is fairly general, but I would certainly have been surprised if it had turned out any other way.

We thank the reviewer for highlighting the link to an experiment reported in Kominsky & Scholl (2020). We report the finding of that experiment now in the revised manuscript.

We added the following paragraph in the discussion:

“For instance, we demonstrated a transfer of adaptation across speed for symmetrical speed ratios. This result complements a previous finding that reported that the adaptation to triggering events (with an asymmetric speed ratio of 1:3) resulted in significant retinotopic adaptation of ambiguous (launching) test events of different speed ratios (i.e., test events with a speed ratio of 1:1 and of 1:3; Kominsky & Scholl, 2020).”

The third experiment tests color (as a marker of object identity), and pits it against motion direction. The results demonstrate that adaptation to red-launching-green generates an adaptation effect for green-launching-red, provided they are moving in roughly the same direction, which provides a nice internal replication of Experiment 1 in addition to showing that the adaptation effect is not sensitive to object identity. This result forms an interesting contrast with the infant causal perception literature. Multiple papers (starting with Leslie & Keeble, 1987) have found that 6-8-month-old infants are sensitive to reversals in causal roles exactly like the ones used in this experiment. The success of adaptation transfer suggests, very clearly, that this sensitivity is not based only on perceptual processing, or at least not on the same processing that we access with this adaptation procedure. It implies that infants may be going beyond the underlying perceptual processes and inferring genuine causal content. This is also not the first time the adaptation paradigm has diverged from infant findings: Kominsky & Scholl (2020) found a divergence with the object speed differences as well, as infants categorize these events based on whether the speed ratio (agent:patient) is physically plausible (Kominsky et al., 2017), while the adaptation effect transfers from physically implausible events to physically plausible ones. This only goes to show that these adaptation effects don't exhaustively capture the mechanisms of early-emerging causal event representation.

We would like to thank the reviewer for highlighting the similarities (and differences) to the seminal study by Leslie and Keeble (1987). We included a discussion with respect to that paper in the revised manuscript. Indeed, that study showed a recovery from habituation to launches after reversal of the launching events. In their study, the reversal condition resulted in a change of two aspects, (1) motion direction and (2) a change of what color is linked to either cause (i.e., agent) or effect (i.e, patient). Our study, based on visual adaptation in adults, suggests that switching the two colors is not necessary for a recovery from the habituation, provided the motion direction is reversed. Importantly, the reversal of the motion direction only affected the perception of causality after adapting to launches (but not to slip events), which is consistent with Leslie and Keeble’s (1987) finding that the effect of a reversal is contingent on habituation/adaptation to a causal relationship (and is not observed for non-causal delayed launches). Based on our findings, we predict that switching colors without changing the event’s motion direction would not result in a recovery from habituation. Obviously, for infants, color may play a more important role for establishing an object identity than it does for adults, which could explain potential differences. We also agree with the reviewer’s point that the adaptation protocol might tap into different mechanisms than revealed by habituation studies in infants (e.g, Kominsky et al., 2017 vs. Kominsky & Scholl, 2020).

We revised the manuscript accordingly when discussing the role of direction selectivity in our study:

“Habituation studies in six-months-old infants also demonstrated that the reversal of a launch resulted in a recovery from habituation to launches (while a non-causal control condition of delayed-launches did not; Leslie & Keeble, 1987). In their study, the reversal of motion direction was accompanied by a reversal of the color assignment to the cause-effectrelationship. In contrast, our findings suggest, that in adults color does not play a major role in the detection of a launch. Future studies should further delineate similarities and differences obtained from adaptation studies in adults and habituation studies in children (e.g., Kominsky et al., 2017; Kominsky & Scholl, 2020).”

One overarching point about the analyses to take into consideration: The authors use a Bayesian psychometric curve-fitting approach to estimate a point of subjective equality (PSE) in different blocks for each individual participant based on a model with strong priors about the shape of the function and its asymptotic endpoints, and this PSE is the primary DV across all of the studies. As discussed in Kominsky & Scholl (2020), this approach has certain limitations, notably that it can generate nonsensical PSEs when confronted with relatively extreme response patterns. The authors mentioned that this happened once in Experiment 3 and that a participant had to be replaced. An alternate approach is simply to measure the proportion of 'pass' reports overall to determine if there is an adaptation effect. I don't think this alternate analysis strategy would greatly change the results of this particular experiment, but it is robust against this kind of self-selection for effects that fit in the bounds specified by the model, and may therefore be worth including in a supplemental section or as part of the repository to better capture the individual variability in this effect.

We largely agree with these points. Indeed, we adopted the non-parametric analysis for a recent series of experiments in which the psychometric curves were more variable (Ohl & Rolfs, Vision Sciences Society Meeting 2024). In the present study, however, the model fits were very convincing. In Figures S1, S2 and S3 we show the model fits for each individual observer and condition on top of the mean proportion of launch reports. The inferential statistics based on the points of subjective equality, therefore, allowed us to report our findings very concisely.

In general, this paper adds further evidence for something like a 'launching' detector in the visual system, but beyond that, it specifies some interesting questions for future work about how exactly such a detector might function.

We thank the reviewer for this positive overall assessment.

**Reviewer #2 (Recommendations for the authors):**
Generally, the paper is great. The questions I raised in the public review don't need to be answered at this time, but they're exciting directions for future work.

We would like to thank the reviewer for the encouraging comments and thoughtful ideas on how to improve the manuscript.

I would have liked to see a little more description of the model parameters in the text of the paper itself just so readers know what assumptions are going into the PSE estimation.

We followed the reviewer’s suggestion and added more information regarding the parameter space (i.e., ranges of possible parameters of the logistic model) that we used for obtaining the model fits.

Specifically, we added the following information in the manuscript:

“For model fitting, we constrained the range of possible estimates for each parameter of the logistic model. The lower asymptote for the proportion of reported launches was constrained to be in the range 0–0.75, and the upper asymptote in the range 0.25–1. The intercept of the logistic model was constrained to be in the range 1–15, and the slope was constrained to be in the range –20 to –1.”

The models provided very good fits as can be appreciated by the fits per individual and experimental condition which we provide in response to the public comments. Please note, that all data and analysis scripts are available at the Open Science Framework (https://osf.io/x947m/).

I also have a recommendation about Figure 1b: Color-code "Feature A", "Feature B", and "Feature C" and match those colors with the object identity/speed/direction text. I get what the figure is trying to convey but to a naive reader there's a lot going on and it's hard to interpret.

We followed the reviewer’s suggestion and revised the visualization accordingly.

If you have space, figures showing the adaptation and corresponding test events for each experimental manipulation would also be great, particularly since the naming scheme of the conditions is (necessarily) not entirely consistent across experiments. It would be a lot of little figures, I know, but to people who haven't spent as long staring at these displays as we have, they're hard to envision based on description alone.

We followed the reviewer’s recommendation and added a visualization of the adaptor and the test events for the different experiments in Figure 2.

**Reviewer #3 (Public Review):**

We thank the reviewer for their thoughtful comments, which we carefully addressed to improve the revised manuscript.

Summary:This paper presents evidence from three behavioral experiments that causal impressions of "launching events", in which one object is perceived to cause another object to move, depending on motion direction-selective processing. Specifically, the work uses an adaptation paradigm (Rolfs et al., 2013), presenting repetitive patterns of events matching certain features to a single retinal location, then measuring subsequent perceptual reports of a test display in which the degree of overlap between two discs was varied, and participants could respond "launch" or "pass". The three experiments report results of adapting to motion direction, motion speed, and "object identity", and examine how the psychometric curves for causal reports shift in these conditions depending on the similarity of the adapter and test. While causality reports in the test display were selective for motion direction (Experiment 1), they were not selective for adapter-test speed differences (Experiment 2) nor for changes in object identity induced via color swap (Experiment 3). These results support the notion that causal perception is computed (in part) at relatively early stages of sensory processing, possibly even independently of or prior to computations of object identity.Strengths:The setup of the research question and hypotheses is exceptional. The experiments are carefully performed (appropriate equipment, and careful control of eye movements). The slip adaptor is a really nice control condition and effectively mitigates the need to control motion direction with a drifting grating or similar. Participants were measured with sufficient precision, and a power curve analysis was conducted to determine the sample size. Data analysis and statistical quantification are appropriate. Data and analysis code are shared on publication, in keeping with open science principles. The paper is concise and well-written.Weaknesses:The biggest uncertainty I have in interpreting the results is the relationship between the task and the assumption that the results tell us about causality impressions. The experimental logic assumes that "pass" reports are always non-causal impressions and "launch" reports are always causal impressions. This logic is inherited from Rolfs et al (2013) and Kominsky & Scholl (2020), who assert rather than measure this. However, other evidence suggests that this assumption might not be solid (Bechlivanidis et al., 2019). Specifically, "[our experiments] reveal strong causal impressions upon first encounter with collision-like sequences that the literature typically labels "non-causal"" (Bechlivanidis et al., 2019) -- including a condition that is similar to the current "pass". It is therefore possible that participants' "pass" reports could also involve causal experiences.

We agree with the reviewer that our study assumes that the launch-pass dichotomy can be mapped onto a dimension of causal to non-causal impressions. Please note that the choice for this launch-pass task format was intentional. We consider it an advantage that subjects do not have to report causal vs non-causal impressions directly, as it allows us to avoid the oftencriticized decision biases that come with asking participants about their causal impression (Joynson, 1971; for a discussion see Choi & Scholl, 2006). This comes obviously at the cost that participants did not directly report their causal impression in our experiments. There is however evidence that increasing overlap between the discs monotonically decreases the causal impression when directly asking participants to report their causal impression (Scholl & Nakayama, 2004). We believe, therefore, that the assumption of mapping between launchesto-passes and causal-to-noncausal is well-justified. At the same time, the expressed concern emphasizes the need to develop further, possibly implicit measure for causal impressions (see Völter & Huber, 2021).

However, as pointed out by the reviewer, a recent paper demonstrated that on first encounter participants can have impressions in response to a pass event that are different from clearly non-causal impressions (Bechlivanidis et al., 2019). As demonstrated in the same paper, displaying a canonical launch decreased the impression of causality when seeing pass events in subsequent trials. In our study, participants completed an entire training session before running the main experiments. It is therefore reasonable to expect that participants observed passes as non-causal events given the presence of clear causal references. Nevertheless, we now acknowledge this concern directly in the revised manuscript.

We added the following paragraph to the discussion:

“In our study, we assessed causal perception by asking observers to report whether they observed a launch or a pass in events of varying ambiguity. This method assumes that launches and passes can be mapped onto a dimension that ranges from causal to non-causal impressions. It has been questioned whether pass events are a natural representative of noncausal events: Observers often report high impressions of causality upon first exposure to pass events, which then decreased after seeing a canonical launch (Bechlivanidis, Schlottmann, & Lagnado, 2019). In our study, therefore, participants completed a separate session that included canonical launches before starting the main experiment.”

Furthermore, since the only report options are "launch" or "pass", it is also possible that "launch" reports are not indications of "I experienced a causal event" but rather "I did not experience a pass event". It seems possible to me that different adaptation transfer effects (e.g. selectivity to motion direction, speed, or color-swapping) change the way that participants interpret the task, or the uncertainty of their impression. For example, it could be that adaptation increases the likelihood of experiencing a "pass" event in a direction-selective manner, without changing causal impressions. Increases of "pass" impressions (or at least, uncertainty around what was experienced) would produce a leftward shift in the PSE as reported in Experiment 1, but this does not necessarily mean that experiences of causal events changed. Thus, changes in the PSEs between the conditions in the different experiments may not directly reflect changes in causal impressions. I would like the authors to clarify the extent to which these concerns call their conclusions into question.

Indeed, PSE shifts are subject to cognitive influences and can even be voluntarily shifted (Morgan et al., 2012). We believe that decision biases (e.g., reporting the presence of launch before adaptation vs. reporting the absence of a pass after the adaptation) are unlikely to explain the high specificity of aftereffects observed in the current study. While such aftereffects are very typical of visual processing (Webster, 2015), it is unclear how a mechanism that increase the likelihood of perceiving a pass could account for the retinotopy of adaptation to launches (Rolfs et al., 2013) or the recently reported selective transfer of adaptation for only some causal categories (Kominsky et al., 2020). The latter authors revealed a transfer of adaptation from triggering to launching, but not from entraining events to launching. Based on these arguments, we decided to not include this point in the revised manuscript.

Leaving these concerns aside, I am also left wondering about the functional significance of these specialised mechanisms. Why would direction matter but speed and object identity not? Surely object identity, in particular, should be relevant to real-world interpretations and inputs of these visual routines? Is color simply too weak an identity?

We agree that it would be beneficial to have mechanisms in place that are specific for certain object identities. Overall, our results fit very well to established claims that only spatiotemporal parameters mediate the perception of causality (Michotte, 1963; Leslie, 1984; Scholl & Tremoulet, 2000). We have now explicitly listed these references again in the revised manuscript. It is important to note, that an understanding of a causal relation could suffice to track identity information based purely on spatiotemporal contingencies, neglecting distinguishing surface features.

We revised the manuscript and state:

“Our findings therefore provide additional support for the claim that an event’s spatiotemporal parameters mediate the perception of causality (Michotte, 1963; Leslie, 1984; Scholl & Tremoulet, 2000).”

Moreover, we think our findings of directional selectivity have functional relevance. First, direction-selective detection of collisions allows for an adaptation that occurs separately for each direction. That means that the visual system can calibrate these visual routines for detecting causal interactions in response to real-world statistics that reflect differences in directions. For instance, due to gravity, objects will simply fall to the ground. Causal relation such as launches are likely to be more frequent in horizontal directions, along a stable ground. Second, we think that causal visual events are action-relevant, that is, acting on (potentially) causal events promises an advantage (e.g., avoiding a collision, or quickly catching an object that has been pushed away). The faster we can detect such causal interactions, the faster we can react to them. Direction-selective motion signals are available in the first stages of visual processing. Visual routines that are based on these direction-selective motion signals promise to enable such fast computations. Please note, however, that while our present findings demonstrate direction-selectivity, they do not pinpoint where exactly that visual routine is located. It is quite possible that the visual routine is located higher up in the visual system, relying on a direction-selective population response as input.

We added these points to the discussion of the functional relevance:

“We suggest that at least two functional benefits result from a specialized visual routine for detecting causality. First, a direction-selective detection of launches allows adaptation to occur separately for each direction. That means that the visual system can automatically calibrate the sensitivity of these visual routines in response to real-world statistics. For instance, while falling objects drop vertically towards the ground, causal relations such as launches are common in horizontal directions moving along a stable ground. Second, we think that causal visual events are action-relevant, and the faster we can detect such causal interactions, the faster we can react to them. Direction-selective motion signals are available very early on in the visual system. Visual routines that are based on these direction-selective motion signals may enable faster detection. While our present findings demonstrate direction-selectivity, they do not pinpoint where exactly that visual routine is located. It is possible that the visual routine is located higher up in the visual system (or distributed across multiple levels), relying on a direction-selective population response as input.”

**Reviewer #3 (Recommendations for the authors):**
- The concept of "visual routines" is used without introduction; for a general-interest audience it might be good to include a definition and reference(s) (e.g. Ullman.).

Thank you very much for highlighting that point. We have chosen the term visual routine to emphasize that we locate the part of the mechanism that is affected by the adaptation in our experiments in the visual system, but at the same time it leaves space regarding the extent to which the mechanism further involves mid- and higher-level processes. The term thus has a clear reference to a visual routine by Ullman (1987). We have now addressed what we mean by visual routine, and we also included the reference in the revised manuscript.

We add the following footnote to the introduction:

“We use the term visual routine here to highlight that our adaptation experiments can reveal a causality detection mechanism that resides in the visual system. At the same time, calling it a routine emphasizes similarities with a local, semi-independent operation (e.g., the recognition of familiar motion patterns; see also Ullman, 1987; Cavanagh, Labianca, & Thornton, 2001) that can engage mid- and higher-level processes (e.g., during causal capture, Scholl & Nakayama, 2002; or multisensory integration, Körding et al., 2007).”

- I would appreciate slightly more description of the phenomenology of the WW adaptors: is this Michotte's "entraining" event? Does it look like one disc shunts the other?

The stimulus differs from Michotte's entrainment event in both spatiotemporal parameters and phenomenology. We added videos for the launch, pass and slip events as Supplementary Material.

Moreover, we described the slip event in the methods section:

“In two additional sessions, we presented slip events as adaptors to control that the adaptation was specific for the impression of causality in the launching events. Slip events are designed to match the launching events in as many physical properties as possible while producing a very different, non-causal phenomenology. In slip events, the first peripheral disc also moves towards a stationary disc. In contrast to launching events, however, the first disc passes the stationary disc and stops only when it is adjacent to the opposite edge of the stationary disc. While slip events do not elicit a causal impression, they have the same number of objects and motion onsets, the same motion direction and speed, as well as the same spatial area of the event as launches.”

In the revised manuscript, we added also more information on the slip event in the beginning of the results section. Importantly, the stimulus typically produces the impression of two independent movements and thus serves as a non-causal control condition in our study. Only anecdotally, some observers (not involved in this study) who saw the stimulus spontaneously described their phenomenology of seeing a slip event as a double step or a discus throw.

We added the following description to the results section:

“Moreover, we compared the visual adaptation to launches to a (non-causal) control condition in which we presented slip events as adaptor. In a slip event, the initially moving disc passes completely over the stationary disc, stops immediately on the other side, and then the initially stationary disc begins to move in the same direction without delay. Thus, the two movements are presented consecutively without a temporal gap. This stimulus typically produces the impression of two independent (non-causal) movements.”

- In general more illustrations of the different conditions (similar to Figure 1c but for the different experimental conditions and adaptors) might be helpful for skim readers.

We followed the reviewer’s recommendation and added a visualization of the adaptor and the test events for the different experiments in Figure 2.

- Were the luminances of the red and green balls in experiment 3 matched? Were participants checked for color anomalous vision?

Yes, we checked for color anomalous vision using the color test Tafeln zur Prüfung des Farbensinnes/Farbensehens (Kuchenbecker & Broschmann, 2016). We added that information to the manuscript. The red and green discs were not matched for luminance. We measured the luminance after the experiment (21 cd/m^2^ for the green disc and 6 cd/m^2^ for the red disc). Please note, that the differences in luminance should not pose a problem for the interpretation of the results, as we see a transfer of the adaptation across the two different colors.

We added the following information to the manuscript:

“The red and green discs were not matched for luminance. Measurements obtained after the experiments yielded a luminance of 21 cd/m^2^ for the green disc and 6 cd/m^2^ for the red disc.”

“All observers had normal or corrected-to-normal vision and color vision as assessed using the color test Tafeln zur Prüfung des Farbensinnes/Farbensehens (Kuchenbecker & Broschmann, 2016).”

- Relationship of this work to the paper by Arnold et al., (2015). That paper suggested that some effects of adaptation of launching events could be explained by an adaptation of object shape, not by causality per se. It is superficially difficult to see how one could explain the present results from the perspective of object "squishiness" -- why would this be direction selective? In other words, the present results taken at face value call the "squishiness" explanation into question. The authors could consider an explanation to reconcile these findings in their discussion.

Indeed, the paper by Arnold and colleagues (2014) suggested that a contact-launch adaptor could lead to a squishiness aftereffect—arguing that the object elasticity changed in response to the adaptation. Importantly, the same study found an object-centered adaptation effect rather than a retinotopic adaptation effect. However, the retinotopic nature of the negative aftereffect as used in our study has been repeatedly replicated (for instance Kominsky & Scholl, 2020). Thus, the divergent results of Arnold and colleagues may have resulted from differences in the task (i.e., observers had to judge whether they perceived a soft vs. hard bounce), or the stimuli (i.e., bounces of a disc and a wedge, and the discs moving on a circular trajectory). It would be important to replicate these results first and then determine whether their squishiness effect would be direction-selective as well. We now acknowledge the study by Arnold and colleagues in the discussion:

“The adaptation of causality is spatially specific to the retinotopic coordinates of the adapting stimulus (Kominsky & Scholl, 2020; Rolfs et al., 2013; for an object-centered elasiticity aftereffect using a related stimulus on a circular motion path, see Arnold et al., 2015), suggesting that the detection of causal interactions is implemented locally in visual space.”

- Line 32: "showing that a specialized visual routine for launching events exists even within separate motion direction channels". This doesn't necessarily mean the routine is within each separate direction channel, only that the output of the mechanism depends on the population response over motion direction. The critical motion computation could be quite high level -- e.g. global pattern motion in MST. Please clarify the claim.

We agree with the reviewer, that it is also possible that critical parts of the visual routine could simply use the aggregated population response over motion direction at higher-levels of processing. We acknowledge this possibility in the discussion of the functional relevance of the proposed mechanism and when suggesting that a distributed brain network may contribute to the perception of causality.

We would like to highlight the following two revised paragraphs.

“[…] Second, we think that causal visual events are action-relevant, and the faster we can detect such causal interactions, the faster we can react to them. Direction-selective motion signals are available very early on in the visual system. Visual routines that are based on these direction-selective motion signals may enable faster detection. While our present findings demonstrate direction-selectivity, they do not pinpoint where exactly that visual routine is located. It is possible that the visual routine is located higher up in the visual system (or distributed across multiple levels), relying on a direction-selective population response as input.”

Moreover, when discussing the neurophysiological literature we write:

“Interestingly, single cell recordings in area F5 of the primate brain revealed that motor areas are contributing to the perception of causality (Caggiano et al., 2016; Rolfs, 2016), emphasizing the distributed nature of the computations underlying causal interactions. This finding also stresses that the detection, and the prediction, of causality is essential for processes outside purely sensory systems (e.g., for understanding other’s actions, for navigating, and for avoiding collisions).”

- p. 10 line 30: typo "particual".

Done.

- p. 10 line 37: "This findings rules out (...)" should be singular "This finding rules out (...)".

Done.

- Spelling error throughout: "underly" should be "underlie".

Done.

- p.11 line 29: "emerges fast and automatic" should be "automatically".

Done.